# The Analysis of Micro-Scale Deformation and Fracture of Carbonized Elastomer-Based Composites by In Situ SEM

**DOI:** 10.3390/molecules26030587

**Published:** 2021-01-22

**Authors:** Eugene S. Statnik, Semen D. Ignatyev, Andrey A. Stepashkin, Alexey I. Salimon, Dilyus Chukov, Sergey D. Kaloshkin, Alexander M. Korsunsky

**Affiliations:** 1HSM Lab, Center for Energy Science and Technology, Skoltech, 121205 Moscow, Russia; a.salimon@skoltech.ru (A.I.S.); alexander.korsunsky@eng.ox.ac.uk (A.M.K.); 2Center for Composite Materials, National University of Science and Technology “MISIS”, 119049 Moscow, Russia; ignatyev.s.11@gmail.com (S.D.I.); a.stepashkin@misis.ru (A.A.S.); dil_chukov@mail.ru (D.C.); kaloshkin@misis.ru (S.D.K.); 3MBLEM, Department of Engineering Science, University of Oxford, Oxford OX1 3PJ, UK

**Keywords:** composite materials, carbonized elastomeric matrices, C/SiC fillers, μ-DENT, in situ tensile test, Deben microtest, Tescan Vega 3, NanoScan-4D, digital image correlation (DIC)

## Abstract

Carbonized elastomer-based composites (CECs) possess a number of attractive features in terms of thermomechanical and electromechanical performance, durability in aggressive media and facile net-shape formability, but their relatively low ductility and strength limit their suitability for structural engineering applications. Prospective applications such as structural elements of micro-electro-mechanical systems MEMS can be envisaged since smaller principal dimensions reduce the susceptibility of components to residual stress accumulation during carbonization and to brittle fracture in general. We report the results of in situ in-SEM study of microdeformation and fracture behavior of CECs based on nitrile butadiene rubber (NBR) elastomeric matrices filled with carbon and silicon carbide. Nanostructured carbon composite materials were manufactured via compounding of elastomeric substance with carbon and SiC fillers using mixing rolling mill, vulcanization, and low-temperature carbonization. Double-edge notched tensile (DENT) specimens of vulcanized and carbonized elastomeric composites were subjected to in situ tensile testing in the chamber of the scanning electron microscope (SEM) Tescan Vega 3 using a Deben microtest 1 kN tensile stage. The series of acquired SEM images were analyzed by means of digital image correlation (DIC) using *Ncorr* open-source software to map the spatial distribution of strain. These maps were correlated with finite element modeling (FEM) simulations to refine the values of elastic moduli. Moreover, the elastic moduli were derived from unloading curve nanoindentation hardness measurements carried out using a NanoScan-4D tester and interpreted using the Oliver–Pharr method. Carbonization causes a significant increase of elastic moduli from 0.86 ± 0.07 GPa to 14.12 ± 1.20 GPa for the composite with graphite and carbon black fillers. Nanoindentation measurements yield somewhat lower values, namely, 0.25 ± 0.02 GPa and 9.83 ± 1.10 GPa before and after carbonization, respectively. The analysis of fractography images suggests that crack initiation, growth and propagation may occur both at the notch stress concentrator or relatively far from the notch. Possible causes of such response are discussed, namely, (1) residual stresses introduced by processing; (2) shape and size of fillers; and (3) the emanation and accumulation of gases in composites during carbonization.

## 1. Introduction

The formulation and use of carbonized elastomer-based composites (CECs) developed in the last decade due to the attractive perspective of reasonable cost net-shaped fabrication of carbon components for oil mining submersible pumps, brakes, current collectors, and other applications [1]. High thermomechanical and electromechanical performance and durability in aggressive media of these materials are, however, accompanied by limited ductility that is closely related to the porosity resulting from gas emanation during carbonization.

Brittleness reduces the suitability of CECs for structural engineering applications, promoting the search for functional applications, where mechanical performance does not represent the primary criterion. Use of hybrid microstructures and miniaturization of structural components seem to represent solutions where CECs can maximize the advantages of their performance.

Fuel cell applications offer a good example of a functional application where CECs can offer overall benefit. Significant progress has been made in the development and implementation of one of the types of fuel cells, namely, redox batteries with an electrolyte based on vanadium salts.

The principle of operation of a fuel cell is to convert chemical energy into electrical energy without combustion and the accompanying damage to the environment. Their inherent advantages are high-efficiency and power density, durability, and low maintenance cost [2,3]. There are several main types of fuel cells, such as polymer electrolyte membrane fuel cells (PEMFCs), high-temperature polymer electrolyte membrane fuel cells (HTPEM FCs), solid oxide fuel cells (SOFCs) and alkaline membrane fuel cells (AMFCs), etc. Within this list, PEMFCs are considered to be most promising for research due to their high efficiency [4].

The primary materials for PEMFCs and HTPEM FCs are polybenzimidazole (PBI) [5,6], poly(arylene ether sulfone)s (PES)s [7], poly(arylene thioether)s (PAT)s [8], poly(arylene ether ketone)s (PAEK)s [9], etc. Working in a temperature range of 160 to 180 °C, these materials have shown their best performance with power density values reaching 780 mW/cm^2^.

The efficient operation of the membrane-electrode blocks requires high chemical and electrochemical resistance in combination with high electrical and thermal conductivity. Currently, fine-grained graphite and graphite-filled polymer compositions are widely used for these purposes. Artificial graphite has high chemical resistance and high electrical conductivity, minimizing electrical losses inside the block, while high thermal conductivity ensures heat dissipation and excludes local overheating inside the unit. Along with advantages, these systems have a severe disadvantage—electrochemical corrosion caused by the oxidation of boundaries between crystallites. The oxidation process has a significant effect on the durability of such blocks. Polymer-graphite compositions show high resistance to chemical and electrochemical corrosion, but unfortunately, they do not have high thermal and electrical conductivity. Another drawback of such materials is the heterogeneity of the structure caused by the difficulty to introduce a large number of conductive fillers and to ensure their homogeneous distribution. To obtain high thermal conductivity and electrical conductivity, it is necessary to reach high filling fractions, which dramatically reduces the strength and deformation characteristics of such materials.

Recently, we put forward a paradigm for the promising alternative to PEMFCs; namely, carbonized elastomer-based composites (CECs), the conductive composite materials based on elastomeric matrices filled with various carbon fillers (carbon fibers, graphite, carbon black, shungite, carbon nanotubes and graphene) passed low-temperature carbonization. The technological workflow developed for CECs makes it possible to create highly filled structures containing up to 75 wt% filler with a highly uniform distribution in the elastomeric matrix [10,11]. Simultaneous use of different carbon fillers facilitates the creation of hierarchically structured heat-resistant material with good electrical conductivity.

The use of specific elastomers (nitrile butadiene rubber (NBR), hydrogenated acrylonitrile butadiene rubber (HNBR), fluorinated caoutchouc) as primary composite matrices plays a vital role in forming composite materials resistant to aggressive media such as those used in fuel cells. The introduction of fillers into the elastomer matrix is carried out using proven rubber industry technologies that ensure uniform distribution of filler particles within the matrix. Vulcanization at 160–180 °C yields complex net-shaped articles of high dimensional accuracy. Subsequent low-temperature carbonization at temperatures 320–360 °C significantly improves the elastic modulus and strength of the composite. It has been shown that fabricated CECs have high values of tensile, compressive, and flexural strength compared to well-known thermoplastic-based composites with polysulfone (PSU), polyphenylene sulfide (PPS), or polyetheretherketone (PEEK) matrices [10,11].

Due to the unavoidable release of gases (oxygen, hydrogen, nitrogen, and water vapor) in the core of the composite, the internal porosity is formed. As a result, carbonized composites become brittle, and their elongation at fracture does not exceed 0.4%. However, brittleness does not represent an insurmountable barrier for successful application when miniature articles are considered since smaller principal dimensions, and thinner cross-sections are less susceptible to residual stress accumulation during carbonization, and to brittle fracture, due to smaller volume for strain energy accumulation. Relatively weakly loaded membrane-electrode blocks in fuel cells and structural elements of MEMS are envisaged as prospective applications where brittleness of CECs may be tolerated. Thorough characterization of micro-deformation and fracture behavior of CECs and the effect of carbonization is required to optimize the mechanical performance and to allow the eventual formulation of predictive models for CAD purposes.

The present study addresses the details of the fracture mechanism in CEC materials subjected to tension. A special experimental setup was employed for in situ in-SEM visualizations and mapping of the spatial distribution and localization of strain in the vicinity of stress concentrators in DENT specimens. The strains determined using digital image correlation (DIC) were correlated with the predictions of finite element modeling (FEM) analysis to refine the values of elastic moduli. The values were compared with the data for elastic moduli obtained using Oliver–Pharr analysis of nanoindentation tests. The analysis of fracture surface appearance (fractography) was used to identify the mechanisms of crack initiation, growth and propagation at different extension rates.

We report novel findings concerning the peculiarities of crack initiation and propagation in notched samples of CECs. A relatively simple protocol for in situ SEM observations and DIC analysis of deformation behavior is presented. It is found that brittle CECs tend to generate curved cracks that follow the bands of strain localization.

## 2. Results and Discussion

### 2.1. In Situ Tensile Testing

Mechanical characteristics of CECs extracted from stress–strain curves (Figure 1) are summarized in Table 1. Taking into account the values of Young’s modulus for filler particulates (10–25, 250–650 and 350–450 GPa for graphite, carbon fibers and SiC, respectively), one can conclude that the simple additive rule of the mixture is not valid for elastic modulus and ultimate tensile strength of vulcanized CECs assuming that matrix-filler bonding is likely to be predominant factor governing their elastic and plastic behavior.

On the other hand, vulcanized CECs having been purposefully developed as the particulate reinforced composites of very high filling degree (75%) may reveal ductile (EC-FC-1 and SiC) or brittle character (EC-FC-1) of fracture depending on the nature of fillers. It seems that quasi-uniaxial particulates in EC-FC-1 and SiC promote some ductility while long sharp fibers are likely to introduce internal concentrators facilitating easy crack propagation.

Carbonization at temperatures below 500 °C involves diverse processes unfolding at heating, e.g., dewatering, emanation of volatile light hydrocarbon fractions, oxidative dehydropolycondensation, thermal destruction inside and further in main chains to form radicals, polymerization of radicals, and, therefore, introduces numerous defects and imperfections at the nanometer, supramolecular dimensional level. A gradual emanation of gases creates pores of submicron and micrometer size frequently associated with filler particulates that weaken the effect of reinforcement. Thermal gradients occurring in big cross-sections at relatively high heating or cooling rates may additionally introduce residual stresses affecting the strength of CEC parts at millimeter dimensional level. The interaction of hierarchically different phenomena takes place in real CEC products such as fuel cell membrane-electrode blocks.

As it can be concluded from the data represented in Table 1 and Figure 1, carbonization significantly improves elastic modulus (at one order of magnitude) and almost doubles ultimate tensile strength for EC-FC-1 and SiC composites. The elongation at break is, however, at least 10 times smaller in carbonized CECs samples with DENT concentrators in comparison with vulcanized counterparts making former ones purely brittle. It is worth noting that crack may pass away from the concentrator, which may suggest (a) the distortion of the stress field pattern from the anticipated one due to the presence of residual stresses; (b) the presence of hidden defects.

This observation is additionally supported by the results obtained for carbonized SiC-filled CEC samples at crosshead speed 0.5; 1.0; 2.0; 1.5 mm/min (Figure 2). Ultimate tensile strength significantly varies in the range of 4.48–26.57 MPa showing no clear dependence on crosshead speed, while elongation at break is much less variable and apparently independent on crosshead speed. The crack may be localized in the DENT neck (1.0 and 2.0 mm/min) or may pass away from the concentrator. This random character of microdeformation and fracture behavior pursues to conclude that inhomogeneity of material properties (local flaws or pores and superposition of tensile residual stress) is the key issue to be mitigated in further technological developments.

### 2.2. Fractography Analysis

The macroscopic appearance of fracture surfaces in various prospective is given in Figure 3 for vulcanized and carbonized composites. It can be seen that in contrast to vulcanized samples, which always fracture across the DENT concentrator neck, some carbonized samples show complex crack propagation paths involving broad zones away from the concentrator and shortest path across the DENT concentrator neck.

The structure of crack surfaces, as seen in Figure 4, depends on the nature of the filler. The crack surface of SiC-filled composite contains many elliptic microscopic hollows. Thus, the tensile fracture mechanism in this composite corresponds mainly to the pull-out of reinforcing particulates from the matrix. A number of microcracks in combination with round hollows were observed at the fracture surface of EC-FC-2 composite samples. The crack surfaces of EC-FC-1 samples reveal no pores, holes or microcracks that correlates with the best mechanical performance in the series of CEC samples studied.

### 2.3. Correlation Between DIC and FE Analyses

The correlation of DIC and FE analysis results was carried out for vulcanized EC-FC-1 composite only since: (1) this composite has the highest mechanical performance after carbonization; (2) cracking develops too fast in carbonized composites that make it impossible to capture a sufficient number of SEM images (each requires at least 4 seconds) and to reach the satisfactory quality of SEM images for reliable DIC analysis.

The pattern of experimentally DIC-detected along X-axis displacements and strains in the vicinity of the DENT concentrator is demonstrated in Figure 5 for several points at stress–strain curve obtained from the Deben microtest device. The nominal stress was calculated as the force per cross-sectional area of the ligament, while strain was found as the ratio of sample elongation to the initial distance between device jaws. The pattern of displacements almost exactly corresponds to the predictions of classical fracture mechanics theory, i.e., under tension, an elliptical region appears of rapid variation of the longitudinal displacement, with the major semi-axis parallel to the loading direction. The strain distribution is clearly inhomogeneous, with the highest strain values present at the notch tips and a central region of relatively low strains located in the middle of the DENT neck. The strain field has a shape of a doughnut elongated parallel to the X-axis and the transverse diameter approximately equal to the width of the DENT sample ligament.

Inhomogeneous strain distribution prevents direct calculation of elastic characteristics (Young’s modulus and Poisson’s ratio) from the data on load and strains as it can be easily carried out for un-notched samples. Therefore, a FE simulation of strain field in a dimensionally identical model was performed, and the patterns of strain fields were correlated (carefully calibrated) with the pattern of experimentally determined strains (Figure 6). Varying the values elastic characteristics and assuming that model isotropic material contains no pores, cracks or other defects, it was found that the best correlation (match) of the modeled strain field with the experimental one is reached for Young’s modulus of 1 GPa, Poisson’s ratio of 0.40, the ultimate tensile strength of 10 MPa and density of 1.61 g/cm^3^. It is worth to note that the central spot of relatively low strains could not be reproduced in simulations; however, the presence of star-shaped zones of relatively low strains in modeled samples is also detected in the middle of the DENT neck.

FEM analysis applied in the present body of research should be considered as a helpful tool with some reservations. First, the size of the meshing element must be taken to be at least several times bigger than the size of reinforcing particulates, pores, and other defects to satisfy the condition of averaging and homogenization of elastic moduli. Long duration roll mixing at the compounding stage resulted in an additional diminishing effect on the fillers, reducing the size of particulates down to units of micrometers [11]. Optimal heating rate and holding time during carbonization in combination with relatively small cross-sections promote the evacuation of emanating gases through the outer sample surface and the formation of small (down to units of a micrometer) pores. We believe that both the small size of imperfections and a reasonable choice of mesh element size allow reaching the high fidelity of FEM results and conclusions. One can note from Figure 6 the good correlation between numerical and experimental strain maps after calibration: Young’s modulus was varied to minimize the overall misfit between modeled and experimental strain values under given load and within a selected view field). The specific features in the vicinity of a DENT concentrator tip and in the wide band between concentrators show the identical topology of strain fields throughout the view field. Discrepancies seem to be related to imperfections of sample geometry and residual computational errors in the FEM algorithm.

It is worth noting that the crack pathways in carbonized CESs are curved, as shown in Figure 3, and they lay somewhat away from the shortest line between the tips of DENT stress concentrators. Moreover, the locus of maximum strains (**lms**) in each x-z cross-section, as demonstrated in Figure 6, is also curved and deviates from the shortest line between the tips of DENT concentrators. One can conclude that brittle CECs showing no plasticity and cracking before the yield are prone to initiate the crack at the DENT tip in the direction along the line of strain concentration and passing in the vicinity of **lms**. This is an important guideline for the reliable design of CECs parts in applications.

### 2.4. Nanoindentation Test

Roughness profiles of samples’ surface were checked (Figure 7 and Figure A1) before nanoindentation tests to fulfill the requirements on the indentation depth that must be in accordance with Section 3.5 of ISO 14577 standard at least 20 times bigger than Ra. Ra roughness at the surface of vulcanized and carbonized samples of EC-FC-1 composite is 0.54 ± 0.10 and 0.51 ± 0.07 μm, respectively. Thus, the indentation depth of about 10 μm is a satisfactory value for reliable results.

The statistics of experimentally measure Young’s modulus values for EC-FC-1 composite (Figure 8) returns the average value of 0.25 ± 0.02 GPa for vulcanized samples and 9.83 ± 1.10 GPa for carbonized samples; for the latter, the values of Young’s modulus occur to be quite close to the values of elastic modulus derived from tensile testing.

## 3. Materials and Methods

### 3.1. Composite Sample Preparation

CECs were prepared from three elastomeric compounds (of compositions given in Table 2), which were formulated and obtained to reach 300 phr filling degree (yielding therefore 75 m. %) through the filling of nitrile butadiene NBR caoutchouc matrix (JSC Krasnoyarsk Synthetic Rubber Plant, Krasnoyarsk, Russia) with three types of carbon fillers: fine granular graphite (GMZ, LC “Moscow Electrode Plant”, Moscow, Russia), dispersed carbon black (CB, N-399, LC “Moscow Electrode Plant”, Moscow, Russia) and chopped carbon fibers (CFs, UKN-5000, CJSC Holding Company "Composite", Moscow, Russia). SiC powder (64C, LLC “NPK Ermakhim”, Moscow, Russia) filler was also applied as a filler for comparison.

CEC sample production was conducted in three stages. At the first stage, a crude elastomeric compound is obtained via the mixing of caoutchouc and solid fillers using a rubber mixing laboratory rolling mill (BL-6175-A, Dongguan Baopin Precision Instrument Co., Ltd, Guangdong, China). The second stage was the vulcanization of compounds at 170 °C and 5 MPa pressure for 10 min with the help of the thermal press (AVPM-901, Tesar-Ingeneering Ltd., Saratov, Russia). Vulcanization causes the spatial crosslinking in the compound matrix that results in an increase in its strength and elasticity accompanied by some decrease of plasticity. Composite samples assume their final shape at the vulcanization stage when the rubber compound is pressed into individual purposefully designed molds. The final stage was the carbonization in a muffle furnace (PM-16M, Electropribor LLC, Saint Petersburg, Russia) at a maximum temperature of 340 °C, which brings the properties CECs to the final values in a series of simultaneous thermal-oxidative destruction and secondary radical polymerization reactions. The workflow of the preparation method is illustrated in Figure 9.

Three samples of each compound composition were cut from the 3–4 mm thick rubber plates and tested after vulcanization and carbonization stages separately. CEC samples with DENT geometry were templated and further cut mechanically with a sharp knife. Polylactide (PLA) templates were additively manufactured using 3D printer Ultimaker 2+ (Ultimaker B.V., Utrecht, The Netherlands). The shape and dimensions of CEC samples are shown in Figure 10. The latter was used in the Deben microtest 1 kN tensile stage (Deben UK Ltd., UK) in accordance with the requirements of ASTM E1820-11 [12] test standard.

Microdeformation and fracture behavior of CECs after vulcanization and carbonization was studied using a Deben miniature testing device in the chamber of SEM in tensile mode and separately using a NanoScan-4D hardness tester. The acquired SEM images were processed to compare the results of DIC analysis with FEA simulations. In addition, the microstructure of fracture surfaces was observed to perform fractography analysis.

### 3.2. In-SEM In Situ Tensile Testing

The tensile test was carried out in situ in the chamber of an SEM Tescan Vega 3 (Tescan Company, Brno, Czech Republic) using a Deben microtest 1 kN tensile stage. Testing was conducted at a permanent crosshead speed of 0.2 mm/min for the specimens of all compounds. The crosshead speed was varied (0.2, 0.5, 1.0, 1.5, and 2.0 mm/min) for the samples of carbonized SiC-filled composite to study the impact of tension rate on the elastic and fracture response. The experimental setup is illustrated in Figure 11. Videos of both vulcanized and carbonized compositions fracture process can be found in the Appendix A.

A Deben microtest 1 kN tensile stage was operated using bespoke Python code to synchronize the mechanical loading with the acquisition of SEM images. The acquisition rate of SEM images was 4 seconds per image (1 μs/pixel) using BSE regime with beam intensity of 15.00, beam size of 500.0 nm, depth of focus of 2.676 mm, the working distance of 32.454 mm, chamber vacuum of 0.98 mPa, and column pressure of 0.25 mPa. Specimens were tested at 0°, and 30° tilt positions, with the latter, used to detect cracks at both the upper and side surfaces.

Figure 12 presents the typical load–displacement curves recorded during tensile tests, along with the appearance of samples after fracturing for CEC after vulcanization and carbonization, respectively.

### 3.3. Digital Image Correlation (DIC) and Finite Element Analysis (FEA)

Digital image correlation (DIC) allows obtaining displacement and strain distributions for deforming specimens with subpixel resolution. Popular Matlab-based open-source software *Ncorr* that proved its versatility and robustness [13] was utilized for DIC analysis. The DIC algorithm is based on the finding of the best match between a chosen pixel subset of two states with the further conversion of determining pixel subset center positions to the displacement and ultimately to strains. *Ncorr* code allows defining the regions of interest and applying various parameters such as shape, the density of points, etc., to create correlation subsets, and eventually interpolating the results to form displacement and strain field maps. The pattern quality (the presence and density of surface features) of an image are crucial for the robustness of DIC analysis. There are several methods to improve the pattern quality, with paint sputtering being the most known and simple. In this paper, the particles of fillers were used as markers for the DIC analysis, as illustrated in Figure 13. The results of the DIC analysis are exemplified in Figure 14.

Since the distribution of displacements and strains is intrinsically inhomogeneous in the specimens with DENT concentrators, it was not possible to directly determine the elastic characteristics such as Young’s modulus and Poisson’s ratio for vulcanized and carbonized samples from experimental stress–strain curves and DIC analysis. For this purpose, the FEA of DENT specimens was performed using Autodesk Fusion 360 [14], with the constraints and force application, as indicated in Figure 15. The dimensions of model samples were taken as indicated in Figure 10. For modeling purposes, the material was considered to be fully elastic, homogeneous, and isotropic (no particulates, pores, cracks, or other defects were taken into consideration) with Young’s modulus set equal to the modulus derived from the analysis of experimental stress–strain curves. A further variation of this value as a parameter to match experimental values of strains (determined by means of DIC analysis) was used to refine the values of Young’s modulus for each type of material investigated. The rectangular mesh in the sample plane contained 857 elements that allowed reaching a spatial resolution for strain mapping of 35 µm, and also ensured convergence of mesh refinement.

### 3.4. Fractography Analysis

Fractography is a well-known approach to deduce the underlying physical mechanisms of fracture and to characterize the crack propagation processes via the examination of fracture surface images acquired using different techniques such as SE or optical microscopy [15]. Ductile fracture evolves through the creation of microvoids disintegrating the material at the fracture surface, then their growth and coalescence to larger voids accompanied by intense plastic flow result in the ultimate destruction. Brittle fracture suggests the development of cracks along weak interior interfaces or weak, thin bands within a phase. Fracture surfaces were studied using SEM Tescan Vega 3 at various magnifications. The examples of fracture surface images used for fractography analysis are represented in Figure 16.

### 3.5. Nanoindentation Analysis

The nanoindentation tests were carried out at ambient temperature using a scanning nanohardness NanoScan-4D tester (NanoScan Company, Troitsk, Russia) equipped with a diamond Berkovitch pyramid. The samples of CECs after vulcanization and carbonization stages were serially indented along a 10 mm line with 250 μm step between individual indents. A linear force increment mode was applied: maximum force was set as 1 N, the load time of 1 s, the hold time of 0.5 s, and free unloading time. Typical indentation curves are demonstrated in Figure 17. Processing of “load–displacement” curves was conducted in accordance with Oliver–Pharr method [16] and ISO 14577 standard [17] to distract the estimates for Young’s modulus and hardness.

## 4. Conclusions

Carbonized elastomer-based composites reinforced by carbon and silicon carbide fillers can be easily net-shaped; they are durable in aggressive media, relatively strong, but brittle; therefore, these composites are attractive for applications excluding significant mechanical loads or impacts. Membrane-electrode blocks in fuel cells and MEMS are viewed as prospective fields for CECs requiring, however, a thorough characterization of their micro-deformation and fracture behavior. Double-edge notched tensile concentrators facilitate to limit the size of the field of view for in situ in-SEM tensile testing. This methodological approach was systematically implemented through the creation of a dedicated experimental setup synchronizing Deben microtest 1 kN tensile stage with scanning system of Tescan Vega 3 SEM. The specimens of CECs prepared using multistage technology that involved mixing, vulcanization, and carbonization stage were tested in tension in the chamber of SEM to acquire a series of high-resolution SEM images for further DIC processing.

Patterns of inhomogeneous strain field are required to be correlated with the results of FE simulations (carried out using Autodesk Fusion 360 program in this research) in order to derive elastic characteristics. On the other hand, these characteristics are able to be calculated from nanoindentation force–displacement curves acquired using a NanoScan-4D hardness tester. Fractography analysis of SEM images collected for fracture surfaces helps to highlight particular mechanisms and the role of filler nature in crack initiation, growth and propagation.

CEC filled with graphite filler showed the best overall mechanical performance both in vulcanized and carbonized state—the highest values of elastic modulus and ultimate tensile strength in combination with acceptable and superior ductility in the vulcanized and carbonized state, respectively. Partial replacement of graphite particulates by carbon fibers aggravates both rigidity, strength and ductility in the carbonized state in comparison with the optimal composite. The effect of filling with SiC particulates is controversial in the carbonized state—high strength and low ductility. Nanoindentation testing returns somewhat smaller values of elastic characteristics that may reflect the impact of the defects/porosity/etc.

Cracks connect notch tips directly, passing the DENT neck along the shortest path in vulcanized composites. In contrast, in carbonized composites, cracks can pass the samples along curved trajectories laying away from the DENT concentrator and approximately along the line connecting the points of maximal strains. This may also hint at the presence and a strong influence of residual stresses inherited after the carbonization stage.

The main outcomes of this research study [18] can be summarized as follows:In situ loading accompanied by Digital Image Correlation (DIC) analysis allows strain mapping in complex-shaped sample(s).Matching FEA simulation to the observations allows reliable determination of material’s Young’s modulus.Carbonization leads to a two-fold increase of strength and considerable reduction of ductility.Fractography reveals the mechanisms of crack initiation and propagation that indicate transition from ductile to ductile-brittle cracking.

## Figures and Tables

**Figure 1 molecules-26-00587-f001:**
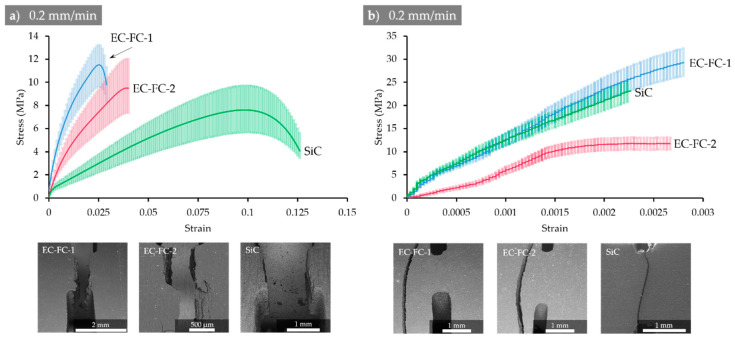
Experimental stress–strain curves with indicated 95% confidence interval for (**a**) vulcanized and (**b**) carbonized specimens that were tested at a constant crosshead speed of 0.2 mm/min. The appearance of broken samples is shown in the lower row of images.

**Figure 2 molecules-26-00587-f002:**
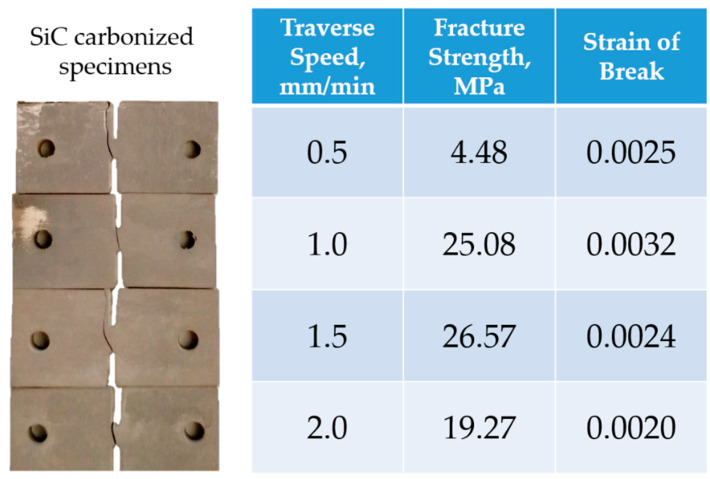
Experimental characteristics obtained for carbonized specimens filled with SiC and tested for 0.5, 1.0, 1.5, and 2.0 mm/min traverse speeds.

**Figure 3 molecules-26-00587-f003:**
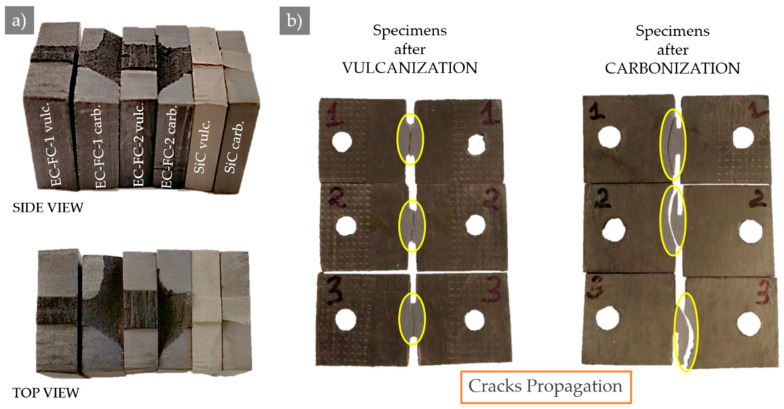
The example of the macro view for different specimens in various perspectives: (**a**) fractography and (**b**) cracks propagation.

**Figure 4 molecules-26-00587-f004:**
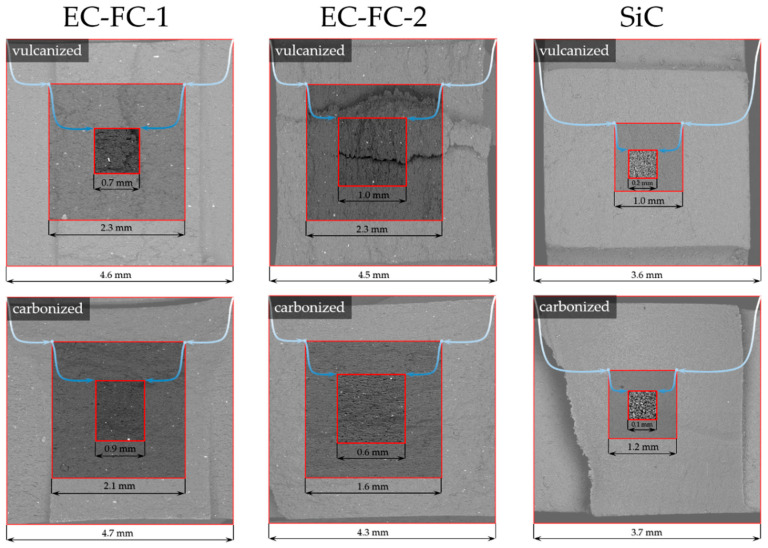
The microscopic appearance of sample fracture surfaces.

**Figure 5 molecules-26-00587-f005:**
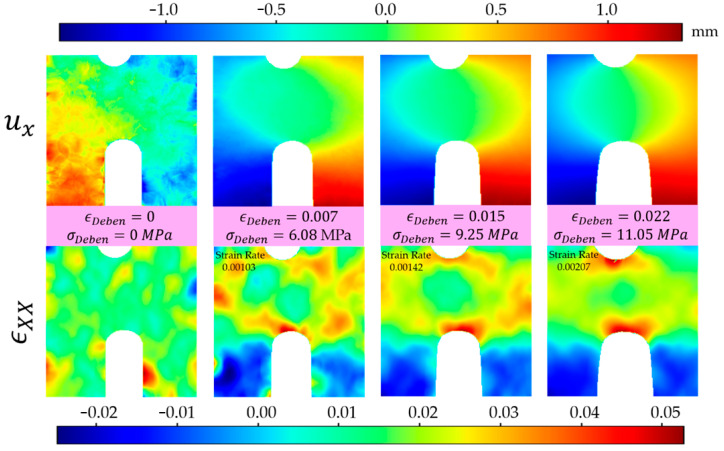
Displacement (**top**) and stain (**below**) distribution along X-axis at nominal macro strain values of 0, 0.007, 0.015 and 0.022, respectively.

**Figure 6 molecules-26-00587-f006:**
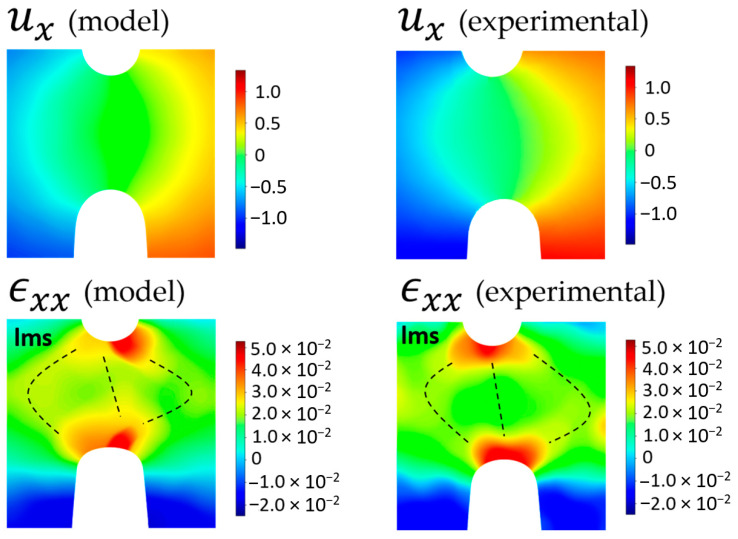
Correlation of modeled (**left**) and experimental (**right**) strain fields.

**Figure 7 molecules-26-00587-f007:**
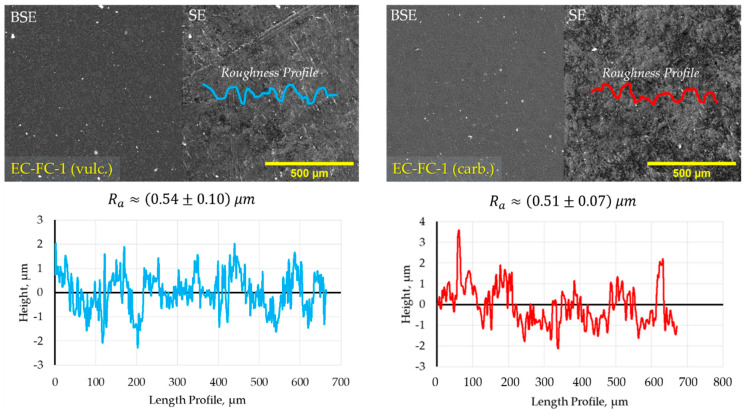
The roughness profile for vulcanized (**left**) and carbonized (**right**) specimens of EC-FC-1 composition.

**Figure 8 molecules-26-00587-f008:**
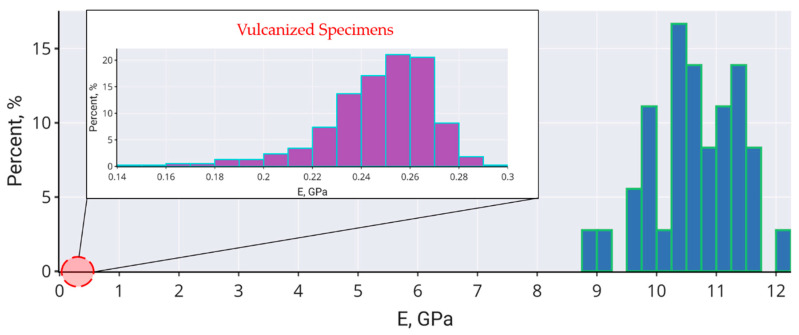
The Young’s modulus distribution of vulcanized (purple) and carbonized (blue) specimens of EC-FC-1 composite.

**Figure 9 molecules-26-00587-f009:**
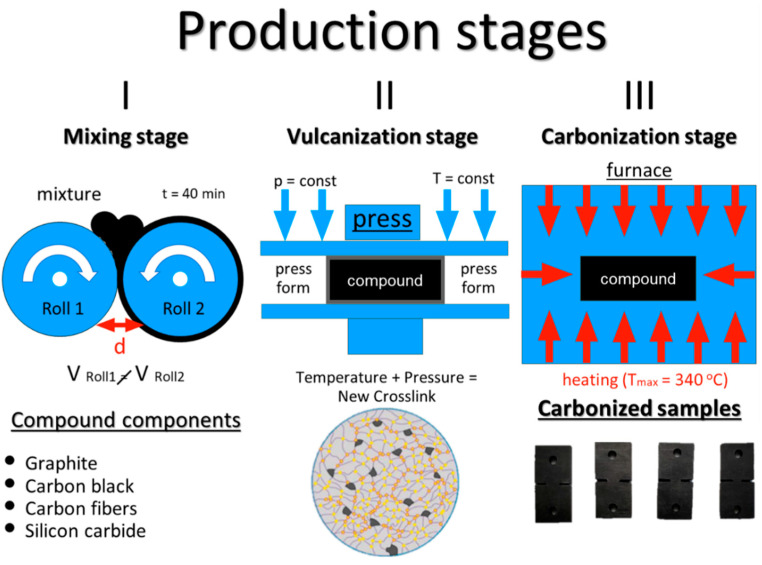
The workflow of preparation composite materials based on the carbonized polymer matrix.

**Figure 10 molecules-26-00587-f010:**
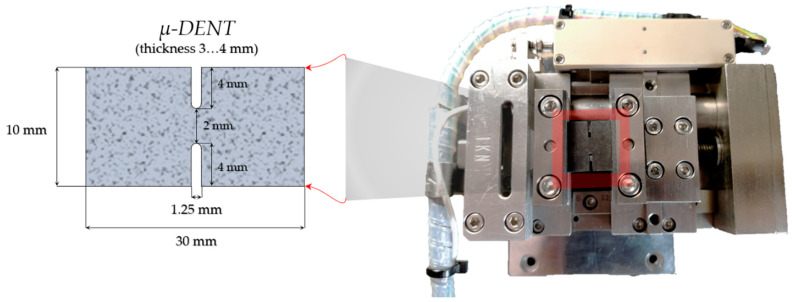
The shape and dimensions of double-edge notched tensile (DENT) specimens.

**Figure 11 molecules-26-00587-f011:**
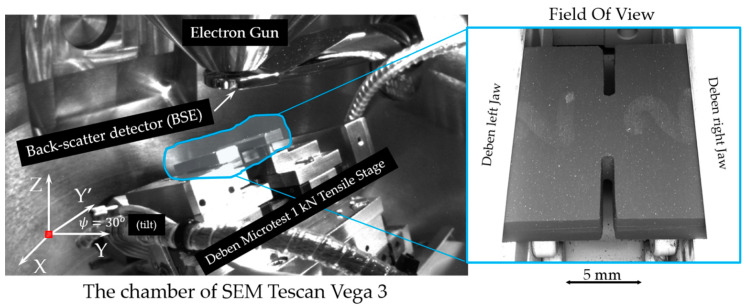
The view of the experimental setup—the chamber of the SEM Tescan Vega 3.

**Figure 12 molecules-26-00587-f012:**
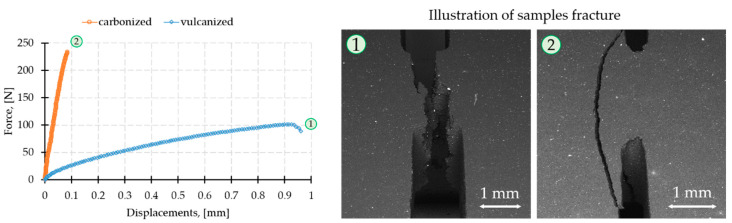
The typical load–displacement curves and fracture appearance for a CEC after vulcanization (**1**) and carbonization (**2**).

**Figure 13 molecules-26-00587-f013:**
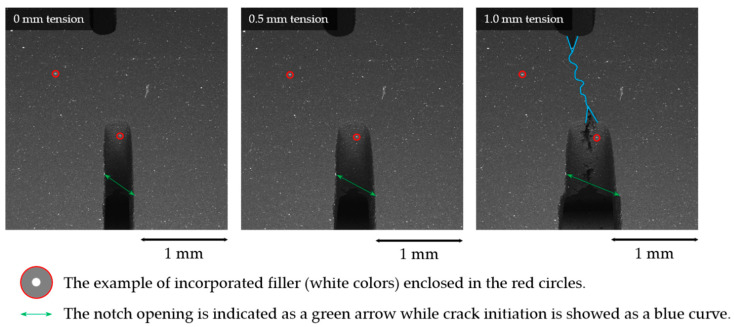
The pattern quality of the captured SEM images.

**Figure 14 molecules-26-00587-f014:**
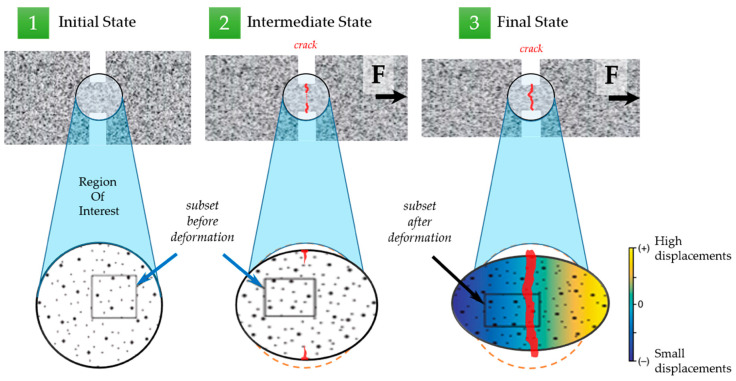
The workflow of the digital image correlation (DIC) analysis.

**Figure 15 molecules-26-00587-f015:**
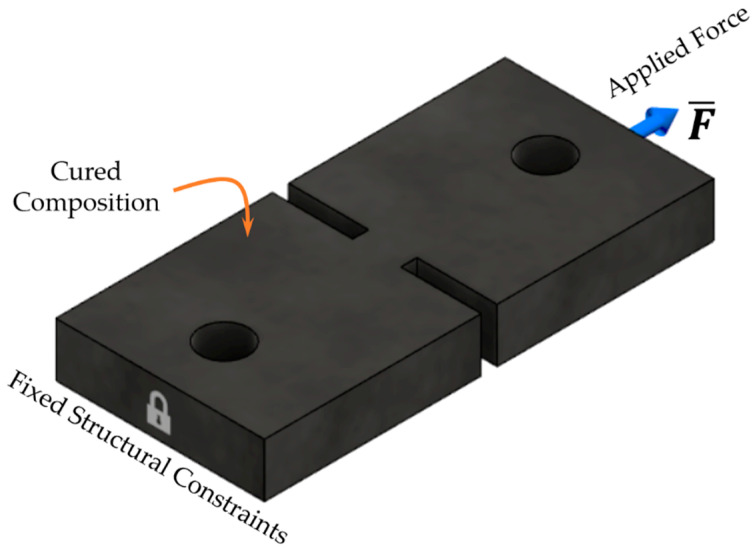
The schematic for finite element analysis (FEA) simulation.

**Figure 16 molecules-26-00587-f016:**
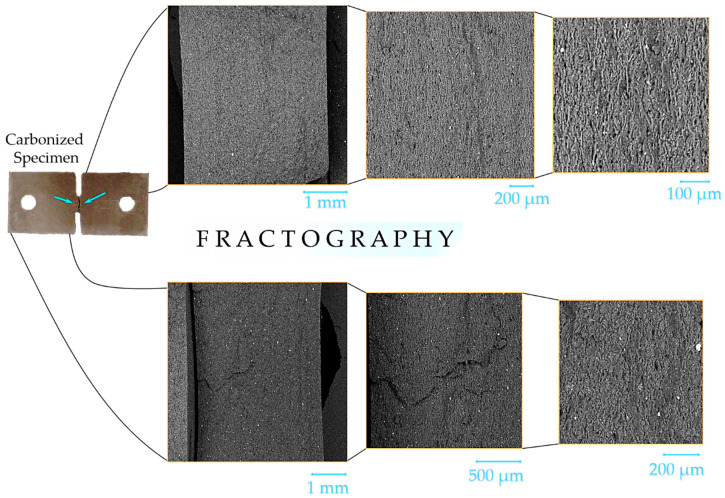
An example of fractography analysis in carbonized elastomer-based composites (CEC) specimens.

**Figure 17 molecules-26-00587-f017:**
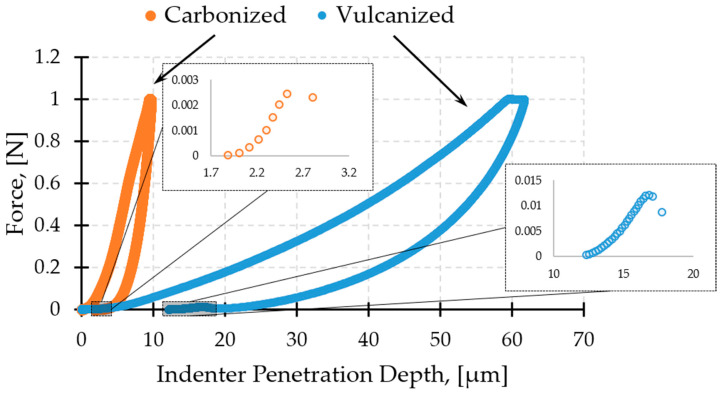
Typical load–displacement curve at nanoindentation.

**Table 1 molecules-26-00587-t001:** Mechanical properties of CECs.

Composite		Elastic Modulus, GPa	Ultimate Tensile Strength, MPa	Strain at Break
EC-FC-1	Vulc.	0.86 ± 0.07	9.80	0.029
	Carb.	14.12 ± 1.20	29.22	0.0028
EC-FC-2	Vulc.	0.44 ± 0.05	9.50	0.040
	Carb.	7.85 ± 1.51	11.75	0.0022
SiC	Vulc.	0.098 ± 0.006	4.07	0.126
	Carb.	13.60 ± 1.34	23.14	0.0026

**Table 2 molecules-26-00587-t002:** Investigated compositions with labels.

Composite	Graphite, phr	Carbon Black, phr	Carbon Fibers, phr	SiC, phr
EC-FC-1	250	50	–	–
EC-FC-2	225	50	25	–
SiC	–	–	–	300

## Data Availability

The data presented in this study are available in the Appendix A.

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
