# Peer review of "The Analysis of Micro-Scale Deformation and Fracture of Carbonized Elastomer-Based Composites by In Situ SEM"

_molecules, 2021, doi:10.3390/molecules26030587_

Round 1

Reviewer 1 Report

Overall, the work is well presented while a significant experimental effort has been made by the authors in order to experimentally characterize carbonized elastomer based composites.

All the data and relevant figures should include the corresponding adopted strain rate, since this parameter is very crucial for the experimental output.

Please give more details about the FEM simulations and modeling. Can the adopted FEM modeling technique treat such inhomogeneous materials? Provide more details about the correlation between experiments and numerical modelling.

In order to validate the followed procedure, the authors should try to provide qualitative comparisons with similar experimental and/or numerical data found elsewhere. There are several studies which cope with similar material components.

Fracture toughness is a key property for characterizing the fracture behavior of a material and perhaps should be included in the results.

Please, clearly define the novelty of the work.

Author Response

Authors would like to thank the reviewers for their detailed comments and suggestions for the manuscript. We believe that the comments have identified important areas which we were able to improve, and that the revised manuscript benefitted in terms of the overall presentation and clarity. Below please find a point-by-point description of how each comment was addressed in the manuscript.

Reviewer 2 Report

The manuscript is really interesting, clear and appropiate to be published in this journal after doing minor changes:

(1) The SEM micrographs (mainly Figure 8); the quality of the images is low. It may be improved in order to clearly see the main fracture mechanisms activated during the in-situ tests.

(2) Furthermore, when the authors epresent the force-indenter penetration depth, it is not possible to distinguish a possible pop-in in the loading and unloading curve. I suggest to reducce the line width and perhaps include a magnification at the contact point.

(3) It will be interesting if the authors can provide to the readers supplemenatry material (video where it is possible to see the in-situ tests). If the authors include that, the quality of the manuscript will increase considerably.

Author Response

(The authors gave the same response as above.)

Round 2

Reviewer 1 Report

The present revised version of the manuscript is appropriate for publication.